# Experimental and Numerical Studies on the Effect of Airflow Separation Suppression on Aerodynamic Performance of a Ducted Coaxial Propeller in Hovering

**Junjie Wang** , **Renliang Chen * and Jiaxin Lu**

Academy of Astronautics, Nanjing University of Aeronautics and Astronautics, Nanjing 210016, China
* Correspondence: crlae@nuaa.edu.cn

**Abstract:** The ducted coaxial propeller (DCP) has great application value in eVTOL aircraft because of its high safety, compactness, and low noise. A numerical simulation method for the DCP is established using the sliding mesh technique. A DCP was designed and manufactured for the lift and power test to verify the numerical method. The characteristics of airflow separation inside the DCP were studied, and the influence of the vortex restrain ring (VRR) on the suppression of airflow separation and on lift augmentation of the duct is analyzed. Results show that, when the tip clearance ratio increases from 0.336% to 1.342%, both the total lift and aerodynamic efficiency decrease by about 11.3%. The influence is mainly reflected in the formation of the tip vortex, airflow separation in the straight, and diffusion sections of the duct. Tip vortex and airflow separation increases DCP energy dissipation and clogs the inner wall of the duct, reducing the effective inner diameter and airflow through the duct. Moreover, the role of the duct is weakened, and the wake is contracted, which increases the induced power loss. By adding a VRR to the diffusion section, the tip vortex and airflow separation can be effectively suppressed, which can increase the aerodynamic efficiency by 5.1%.

**Keywords:** ducted coaxial propeller; computational fluid dynamics; numerical simulation; aerodynamic interference; eVTOL

## 1. Introduction

Urban air mobility (UAM) has received wide attention with the decarburization of modern cities [1–23]. To save time in take-off and landing, reducing dependence on runways, electrically driven vertical take-off and landing (eVTOL) aircraft with an electric distributed propulsion system are among the best choices [4]. The ducted coaxial propeller (DCP) is a lift device with a pair of counter-rotating propellers placed within a duct. In contrast to the free coaxial propeller (FCP), the DCP has larger aerodynamic efficiency, smaller size, higher safety, lower noise, etc. [5], in hovering and forward flight at low speed. In addition, the additional propeller and motor in the DCP make the aircraft more fault tolerant and thus more reliable and safer. Therefore, the DCP has become a popular choice of power system for eVTOL aircraft, and has received a lot of attention from researchers [6,7].

Ensuring excellent aerodynamic characteristics of the DCP in hovering is one of the most important design aims of eVTOL aircraft. However, the internal flow field of the duct has significant unsteady aerodynamic characteristics under joint interaction of propeller tip flow leakage, flow in the boundary layer of the duct wall, and propeller wake. The resulting clogging, drag, and energy loss all interact with each other, making it an important source of energy loss of the DCP. The duct can suppress the propeller tip vortex and thus reduce tip vortex energy loss, so theoretically, a smaller tip clearance is better for DCP design [8]. However, it is difficult to achieve the ideal condition in engineering applications. On the one hand, the speed of the propeller tip exceeds 200 m/s in order to obtain a high propeller disc load, so for safety, an appropriate distance needs to be maintained to strictly avoid

any collision of the rotating propeller tip with the inner wall of the duct. On the other hand, the vibration of the moving components, the centrifugal force of the high-speed rotation of the propeller, and the machining accuracy of the large-sized duct and propeller all need to be taken into consideration in the design and machining processes of the duct and propellers, which brings a significant limitation to the control of tip clearance size. In the meantime, the airflow attached to the inner wall of the duct is prone to separation due to the joint influence of the tip vortex and propeller wake contraction, resulting in a reduction in the effective cross-sectional diameter of the propeller slipstream, which thus degrades aerodynamic performance. This paper aims to study the separation mechanisms of the flow in the surface layer of the inner wall of the duct under the coupling of the tip vortex generated at the tip clearance, wall boundary layer viscosity, and wake contraction, as well as the affecting parameters on aerodynamic performance. The study also proposes the use of the vortex restrain ring (VRR) to suppress the flow separation phenomenon, so as to improve the DCP's aerodynamic efficiency. This paper provides technical guidance for the practical engineering design and application of eVTOL aircraft in improving aerodynamic characteristics.

Current research on the duct fan is mainly focused on experimental studies and computational fluid dynamics *(CFD)* simulations [9]. The experimental research results are highly credible, but they are costly and difficult to implement. Additionally, there are factors, such as experimental model processing errors, measurement accuracy, and ground environment interference, that may affect the experimental results. As for the experimental study of the DCP, on the one hand, the flow field detail changes within the duct cannot be directly observed; on the other hand, the test is difficult to perform because the measurement of the aerodynamics of the two propellers and the duct is difficult to distinguish from each other [10]. The *CFD* calculation method based on the N–S equation can carefully capture the detailed flow in the boundary layer of the duct and the interference between the blades and vortex [11], which has certain advantages in this case and is suitable for the simulation of DCP flow fields with complex flow mechanisms [12,13]. Biava, M. et al. studied the effects of propeller position, duct shape, propeller twist angle, and tip clearance on the aerodynamic performance of the DSP using the *CFD* method, and improved aerodynamic efficiency at low flight speed by rational optimization [14–16]. Singh, R. et al. studied the aerodynamic interference between DSPs by combining *CFD* calculations and experiments [17–19], which could both improve the confidence level of the study and obtain a detailed interference flow field. Current studies mainly focus on the DSP, and there is not much elaboration and analysis on the effect of DCP tip clearance on the mutual interference between the propeller and the duct, as well as on the corresponding airflow separation mechanisms. Therefore, it is still necessary to perform more in-depth research on this. Moreover, most research in the related literature is on small duct fans, while research on large ducted fans for eVTOL aircraft is scarce. However, the two kinds of fans have different operation Reynolds numbers; thus, they definitely have different aerodynamic characteristics. Moreover, research on the improvement of aerodynamic characteristics is mainly focused on the geometric parameters of the duct cross-sectional profile, while less research work has been performed on increasing the tiny VRR on the inner surface of the duct.

In this paper, we designed and fabricated a set of DCPs with a diameter of 1.78 m, and established a set of experimental systems for aerodynamic testing of the DCPs, which can measure the lift and power of the DCP in hovering. However, it is difficult to measure the aerodynamic forces and capture the tip vortex and airflow separation flow field details between different components separately in the experiment, so another numerical simulation method set based on the slip grid model and SST k-w turbulence model was established, and the confidence of the *CFD* method was verified with the experimental data. The rest of this paper is constructed as follows. In Section 2, the research object, grid refinement, and applicable numerical simulation method are introduced. In Section 3, the lift and power test for the DCP is introduced and carried out to verify the simulation

method; time-step sensitivity is also performed. Section 4 investigates the influence of tip clearance on the aerodynamics and the phenomenon and mechanisms of airflow separation inside the DCP; in addition, suppression of airflow separation by adding the VRR is also studied and discussed. Some conclusions are presented in Section 5.

## 2. Methods

### 2.1. System Description

A novel eVTOL aircraft using the DCP lift system was designed, as shown in Figure 1. Four DCPs in total, each connected to the fuselage, were used showing an X-shaped aerodynamic layout. The DCP was preliminarily designed in the upper and lower propellers based on momentum element theory targeting hovering efficiency, with main parameters such as chord length and torque distribution determined first, and then those parameters were optimized through *CFD* calculation. The DCPs, mainly for hovering state and forward flight at low speed, were designed with 3 blades on each propeller. DCPs have no superior efficiency in high-speed forward flight, so this condition is not studied in this paper [21].

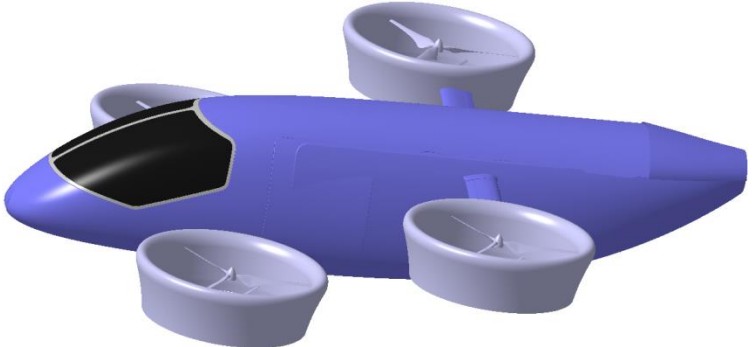

**Figure 1.** eVTOL aircraft with DCP rendering.

This paper focuses on the aerodynamic characteristics of the DCP in hovering. So, for convenience, the model is simplified by removing the motor, connection structure, and other components that have little influence on aerodynamic characteristics. The simplified model, consisting of an upper propeller, a lower propeller, and a duct, is shown in Figure 2. Its main parameters are shown in Table 1. The internal part of the duct wall consisted of lip, line segment, and diffuser, and the whole duct structure is shown in Figure 2a. The airfoil profile of the duct is shown in Figure 2c. The propeller discs and vortex restrain ring are within the line segment region, and the duct lip is at the inlet. The distance between the inner wall of the duct and the propeller tips is 5 mm.

**Table 1.** Main parameters of DCP.

| Parameter | Value |
| --- | --- |
| Propeller diameter, m | 1.49 |
| Propeller airfoil | CLARK-Y |
| Spacing between propellers, m | 0.32 |
| Chord of duct, m | 0.56 |
| Inner diameter of duct, m | 1.5 |
| Outer diameter of duct, m | 1.78 |
| Tip clearance, m | 0.005 |
| Design, RPM | 3000 |

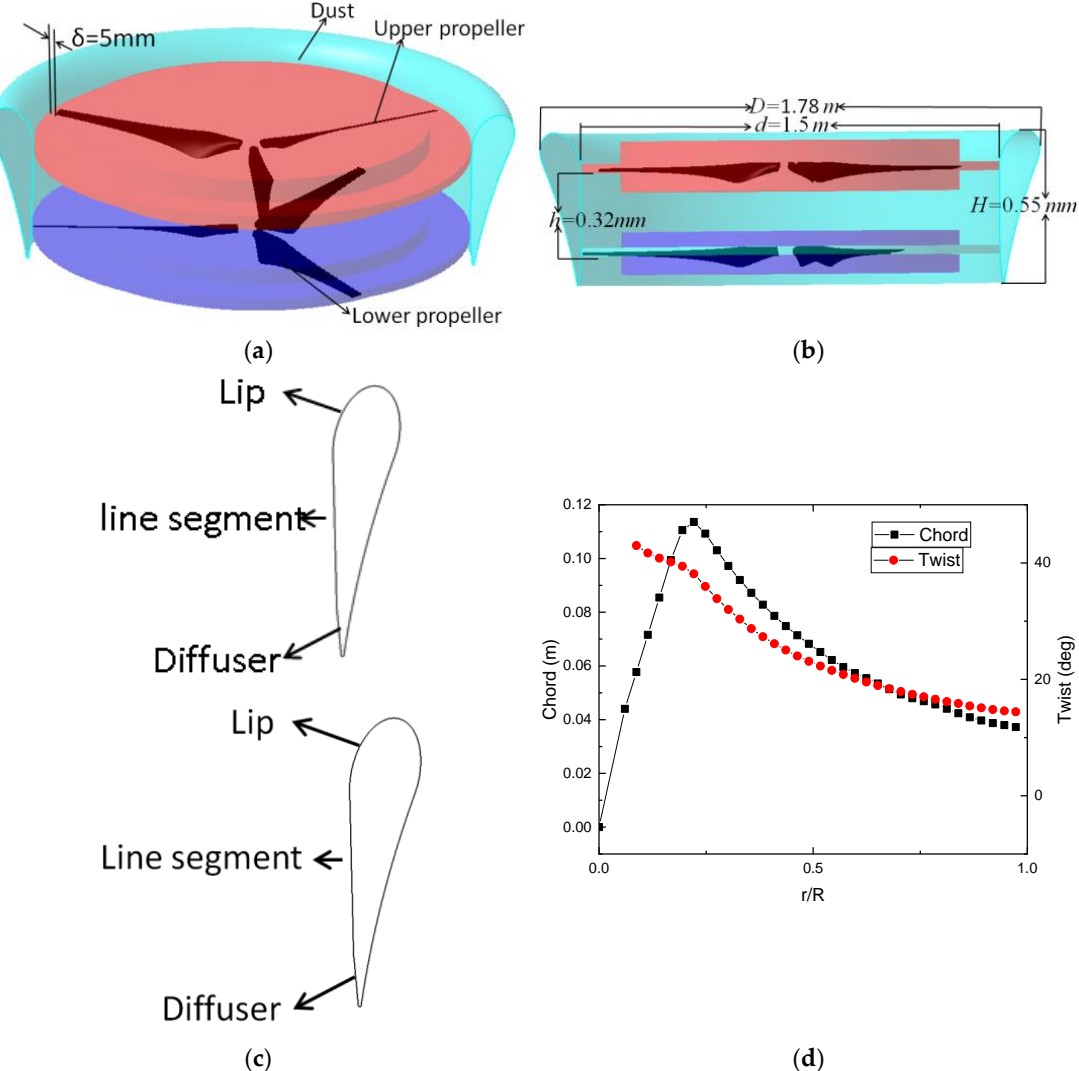

**Figure 2.** Introduction of DCP parameters. (**a**) parts diagram; (**b**) main parameters; (**c**) duct airfoil; (**d**) propeller chord and twist.

### 2.2. Grid Refinement

Grid quality determines *CFD* calculation accuracy. Since the research object involves high-speed rotation in a low-clearance environment, to accurately simulate the viscous and turbulent flow field around and inside the DCP, the difficulty lies in how to choose the appropriate outer domain meshing method and boundary layer meshing method, correctly deal with the flow field boundary conditions, and control the number of meshes to improve the calculation efficiency. The DCP benchmark model has a tip clearance of only 5 mm, which makes it very easy to have a negative volume in the process of mesh division, bringing more difficulty to the research. A grid with an unstructured mesh is established for the DCP system using the commercial CFD software ICEM(ANSYS 18.0). For the boundary conditions, the inlet is set as the velocity inlet, and the outlet is set as the pressure outlet, as shown in Figure 3. The overall computational domain is set as a cylinder. In order to better simulate the flow field around the DCP, the small cylinders generated around the DCP and the intersection interface are refined in meshing to improve computational efficiency. The sliding mesh technique is used to simulate propeller rotation, which not only helps to avoid distortion of the surface mesh of the boundary layers, but also improves efficiency. The fluid domain is divided into one static domain and two dynamic domains, and the duct is located in the static domain, which surrounds the two

dynamic zones. The grid of the rotational domain is further refined so that the mesh there has a good quality. For the duct and the propellers, a prismatic mesh is used for boundary layer meshing to produce a better viscous boundary layer mesh to pounce on the better near-wall flow, and the computational domain is filled by tetrahedral cells.

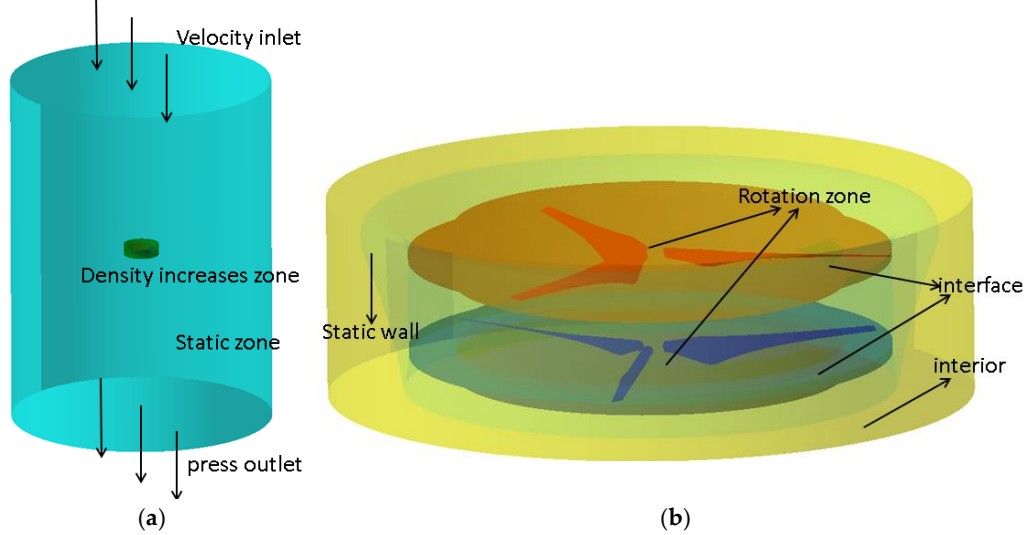

**Figure 3.** Boundary conditions: (**a**) static zone; (**b**) rotation zone.

The Reynolds number of the flow is in the range of $10^5$–$10^6$. The flow around both the propeller and duct is viscous flows, and there is a more complex flow structure in the boundary layer, which requires a proper mesh of the boundary layer to better simulate the viscous interaction within the boundary layer and the complex flow structure after the interference between the propeller and duct. The flow field around the two has different airflow velocities; thus, different boundary layer thicknesses should be used. The duct has a small surface velocity and chord length, while the propeller has a large tip velocity and a small chord length. Therefore, they can share the same height in the first layer of the boundary layers. The grid is refined step by step. The grid height (d1) of the first boundary layer grid is set to 0.0001 m, 0.00005 m, 0.00003 m, and 0.00001 m, the growth rate is set to 1.2, and the total number of layers is 12. The total lift of the DCP in hovering is calculated when the speed of both the upper and lower propellers is at 2500 RPM. The results of the mesh irrelevance analysis are given in Figure 4 shows. The iteration error of the calculation is shown in Formula (1), where $L_{Newmesh}$ is the lift using the new grid, and $L_{Oldmesh}$ is the lift using the last grid. The results reveal that the grid with d1 = 0.00003 m achieves iteration error smaller than 0.6%, which can be considered as having no obvious influence on calculation results.

$$Iteration\_error = \frac{L_{Newmesh} - L_{Oldmesh}}{L_{Newmesh}} \times 100\% \tag{1}$$

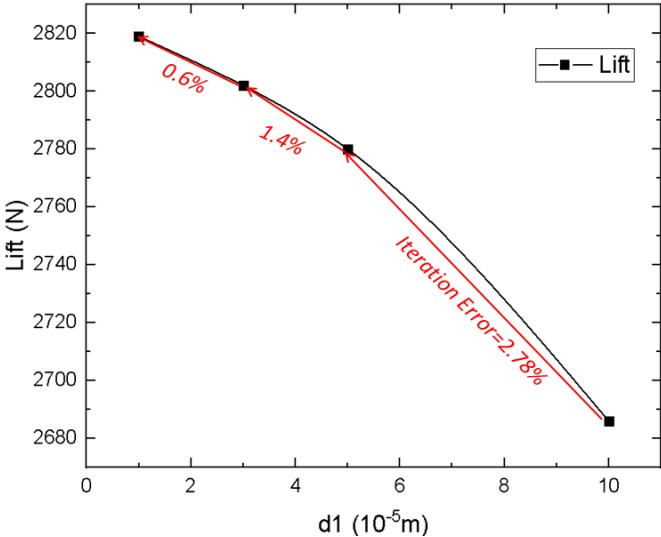

**Figure 4.** Lift of DCP with different heights of the first boundary layer.

To study the effect of grid density on the calculation results, the grids are deliberately refined in the dynamic domains enclosing the propellers and the static domain enclosing the duct. The total number of grid cells is 6 million, 10 million, 14 million, and 18 million. The total lift of the DCP in hovering is calculated when the speed of both the upper and lower propellers is at 2500 RPM, and the results are shown in Figure 5. Considering the accuracy and economy of the calculation, 14 million cells were chosen to calculate the aerodynamic performance of the model in hovering.

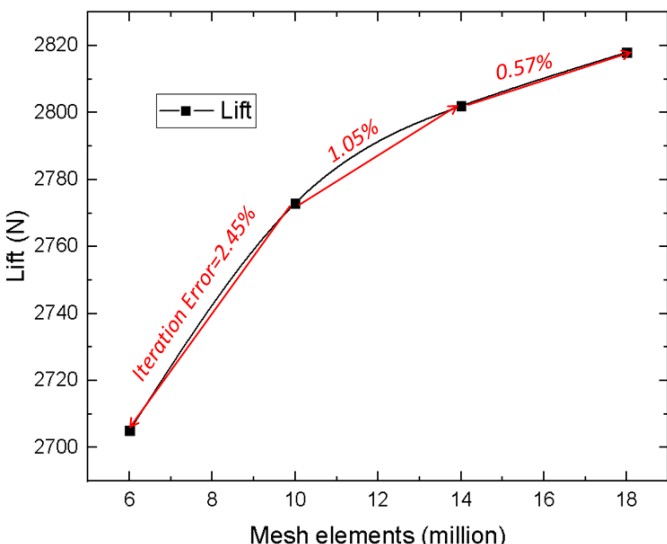

**Figure 5.** Lift of DCP with different mesh density.

According to the grid irrelevance analysis, a grid with 14 million cells and the height of the first layer of boundary layers of 0.00003 m is enough for accurate calculation. The final number of boundary layer layers of both the duct and the propeller is 12, the wall growth rate of the boundary layer grid is 1.2, and the value of the grid wall y+ is 1. The grid of the DCP is shown in Figure 6.

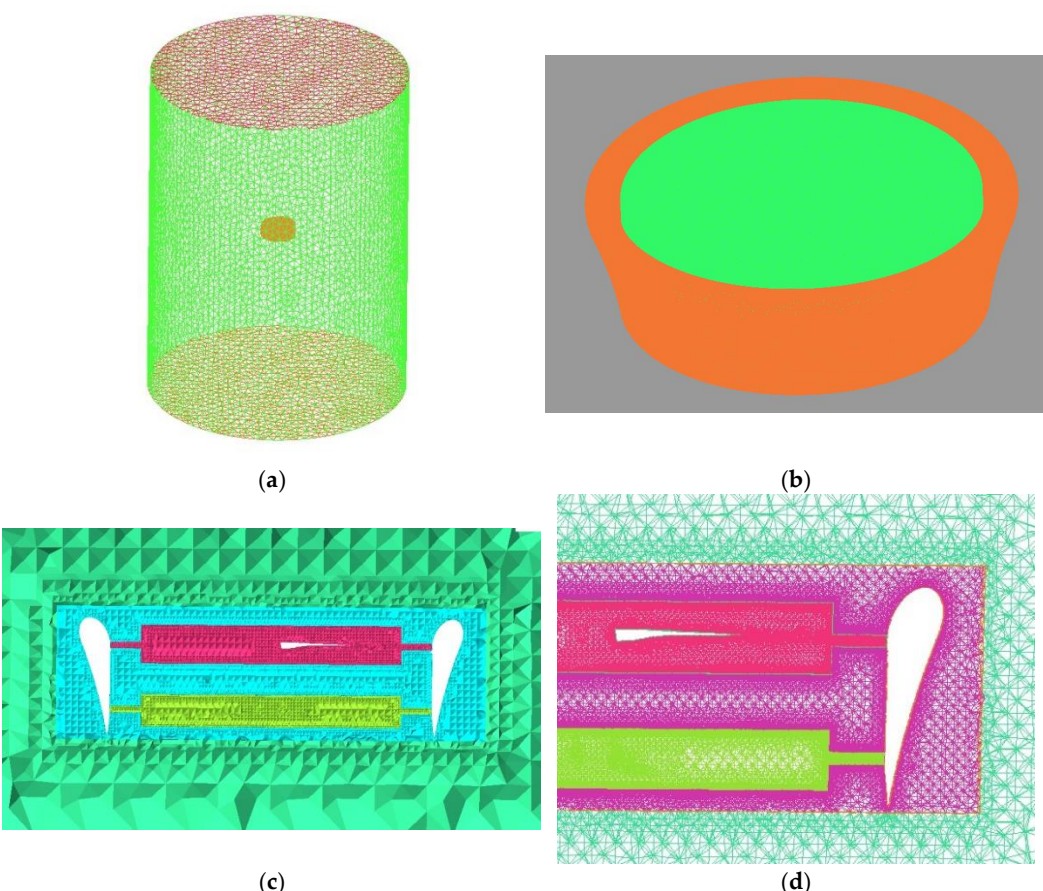

**Figure 6.** Grid diagram: (**a**) far-field grid; (**b**) grid of duct surface; (**c**) body grid sections; (**d**) grid detail of propeller and duct.

*2.3. Numerical Methods*

Commercial *CFD* software FLUENT (ANSYS 18.0) was used to model the three-dimensional flow structure of the DCP, and numerical simulations on unsteady flow were performed to analyze the viscous and turbulent flow fields around and inside the DCP under hovering conditions, especially for the complex flow fields around the blades and near the interior of the DCP, which needs accurate mathematical model for prediction. The rotational speed for calculation is 2500 rpm, with a corresponding Mach number larger than 0.3 Ma; therefore, the air compressibility needs to be taken into consideration. The Navier–Stokes equation is solved for the entire computational domain using the governing equation, with the expression form as:

$$\frac{\partial}{\partial t} \int_{\Omega} W d\Omega + \int_{\partial\Omega} (F_c - F_v) ds = \int_{\Omega} Q d\Omega \tag{2}$$

where $W$ is conservative variables, $F_c$ is the vector of convective fluxes, $F_v$ is the Vector of viscous fluxes, $Q$ is the source term, $t$ is the time, $\Omega$ is the control volume, $d\Omega$ is the boundary of a control volume, $ds$ is the surface element.

For the vector of conservative variables, we obtain:

$$W = \begin{bmatrix} \rho & \rho u & \rho v & \rho w & \rho E \end{bmatrix}^T \tag{3}$$

where $\rho$ is the density, $E$ is the total energy per unit mass.

For the vector of convective fluxes, we obtain:

$$F_c = \begin{bmatrix} \rho V \\ \rho u V + n_x p \\ \rho u V + n_y p \\ \rho w V + n_z p \\ \rho H V \end{bmatrix} \tag{4}$$

where $p$ is the static pressure, H is total enthalpy, $V = V \cdot n = n_x u + n_y v + n_z w$, the velocity vector $V = \begin{bmatrix} u & v & w \end{bmatrix}^T$, the unit normal vector $n = \begin{bmatrix} n_x & n_y & n_z \end{bmatrix}^T$.

For the vector of viscous fluxes we have with $F_v$:

$$F_v = \begin{bmatrix} 0 \\ n_x \tau_{xx} + n_y \tau_{xy} + n_z \tau_{xz} \\ n_x \tau_{yx} + n_y \tau_{yy} + n_z \tau_{yz} \\ n_x \tau_{zx} + n_y \tau_{zy} + n_z \tau_{zz} \\ n_x \Theta_x + n_y \Theta_y + n_z \Theta_z \end{bmatrix} \tag{5}$$

where:

$$\begin{aligned} \Theta_x &= n_x \tau_{xx} + n_y \tau_{xy} + n_z \tau_{xz} + k\frac{\partial T}{\partial x} \\ \Theta_x &= n_x \tau_{yx} + n_y \tau_{yy} + n_z \tau_{yz} + k\frac{\partial T}{\partial y} \\ \Theta_x &= n_x \tau_{zx} + n_y \tau_{zy} + n_z \tau_{zz} + k\frac{\partial T}{\partial z} \end{aligned} \tag{6}$$

$$\tau_{ij} = \mu\left(\frac{\partial \mu_i}{\partial x_j} + \frac{\partial \mu_j}{\partial x_i}\right) + \lambda(\nabla \cdot V)\delta_{ij}, \lambda + \frac{2}{3}\mu = 0 \tag{7}$$

where $\tau_{ij}$ is the components of viscous stress tensor, $\mu$ is the dynamic viscosity coefficient, and $k$ is the thermal diffusivity coefficient.

In order to accurately simulate the tip vortex, wake flow and airflow separation after mutual induction and distortion inside the duct, the $k$-$\omega$ SST (shear stress transport) turbulence model is chosen [21,22]. The $k$-$\omega$ model can accurately simulate the bottom flow in the boundary layer of the DCP and the unsteady flow field inside the duct, with better stability and adaptability for complex flow and compressible flow [23]. The turbulence kinetic energy equation is:

$$\frac{\partial}{\partial t}(\rho \kappa) + \frac{\partial}{\partial x_i}(\rho \kappa u_i) = \frac{\partial}{\partial x_i}\left(\Gamma_k \frac{\partial k}{\partial x_j}\right) + G_k + Y_k \tag{8}$$

$$\frac{\partial}{\partial t}(\rho \omega) + \frac{\partial}{\partial x_j}(\rho \omega u_j) = \frac{\partial}{\partial x_j}\left(\Gamma_\omega \frac{\partial \omega}{\partial x_j}\right) + G_\omega - Y_\omega + D_\omega \tag{9}$$

where, $G_k$ and $G_\omega$ represent the turbulent kinetic energy term $k$ and the specific dissipation rate term $\omega$ generated by the mean velocity gradient; $u_i$ and $u_j$ are the time-averaged velocities, $\Gamma_k$ and $\Gamma_\omega$ are the effective diffusion terms of $k$ and $\omega$, respectively; $D_\omega$ is the cross-diffusion term, The dissipation term of $k$ is represented by the expression $Y_k$. The dissipation term of $\omega$ is represented by the expression $Y_\omega$. The flow velocity distribution in the boundary layer calculated using the $k$-$\omega$ SST turbulence model is shown in Figure 7. It can be seen that the velocity along the rotor wall's normal direction grows from 0 m/s to 80 m/s, and the change trend is consistent with the boundary layer theory, which proves the rationality of using the SST turbulence model.

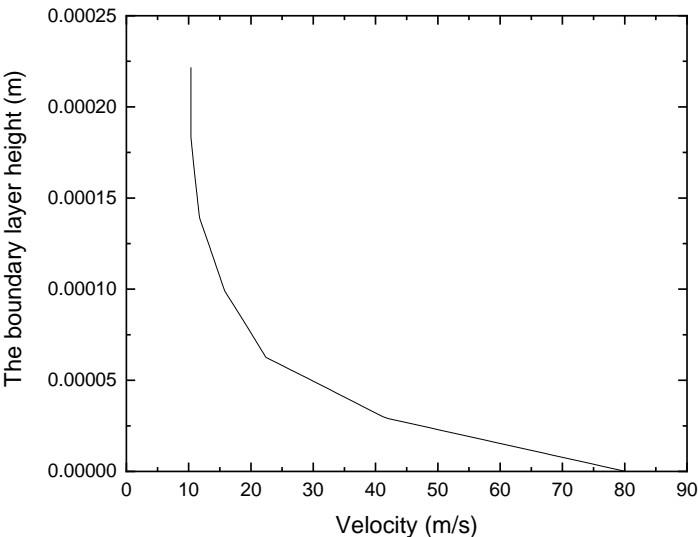

**Figure 7.** Boundary layer velocity distribution.

The numerical methods for solving the Navier–Stokes equations are discretized separately in space and time. The finite volume method with second-order accuracy is used for spatial discretization [24]. The ROE format is used to calculate the convective flow, and the approximate Riemann approximation is used to solve the convective flow on the grid boundary [25]. The velocity and pressure coupled solver is used to solve the continuum, dynamical, and energy equations simultaneously. In order to improve the efficiency of solving the unsteady flow field, the dual-time implicit lower upper symmetric Gauss–Seidel (LU-SGS), the iterative method is used for the time discretization [26]. The intersection surfaces of the rotation domains adopt circumferential flux-conserving connection surfaces; namely, in the surfaces mass, momentum and energy are strictly conserved.

### 3. Experiment and Validation

In order to verify the numerical simulation method used in this paper, an electric large DCP was designed and fabricated. Tests on lift and power measurement were performed on this DCP system.

### 3.1. Experiment Setup

The experimental rig, as shown in Figure 8, consists of four parts: a DCP system, a measurement system, a power system, and an auxiliary system. The DCP system consists of 2 emrax 228 motors, 2 automotive ESCs, a cooling system, an external duct, and 2 propellers of 1.49 m. The measurement system is mainly installed at the bottom of the inner frame of the tensile pressure sensor and has a measurement result error smaller than 0.3%. The measured data are displayed through a digital data display system in real time. The external duct is fixed on the main stand using a threaded round bar, the center stand supports the DCP system in a way that the DCP is 0.3 m above the ground, and the DCP system outlet is facing outward. The motor is powered by a 450 v DC power supply cabinet and the motor is controlled by an electronic stability controller (ESC). The motor itself is capable of continuously monitoring rotational speed, current, and temperature. The experimental principle is to simulate the hovering state of the DCP with the ground experiment by installing the DCP system on the fixed base, using the motor to drive the two propellers for rotation, controlling the current through the ESC to achieve different rotational speeds, and then reading the pressure value according to the pressure sensor to obtain the value of the lift generated by the DCP system. At the same time, the voltage $U$ and current $I$ values of the power supply cabinet are recorded to obtain the power consumption. This experimental system is used to simulate the change in lift and power consumption of the DCP in hovering at different rotational speeds. The propeller is placed vertically

(above ground at a certain height) as opposed to horizontally to eliminate most of the ground effect lift, so as to make the results closer to the real situation. The experiments measured the lift, voltage, and current of the DCP at speeds from 500 rpm to 2500 rpm, and finally, the experimental values of lift and power consumption generated by the propeller were obtained.

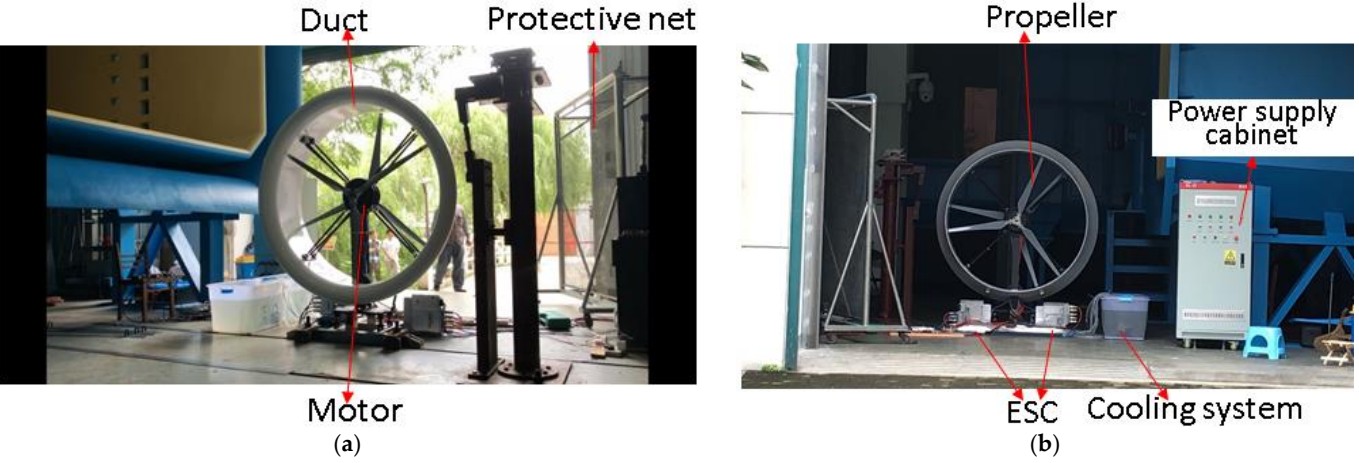

**Figure 8.** Lift and power measurement test bench: (**a**) duct inlet; (**b**) duct outlet.

Test irrelevance analysis: (1) The experimental main bracket in the middle of the DCP has a certain augmentation effect on the experimental results. The projected area of the test main bracket column along the propeller plane, which will block a small part of the airflow and produce a ground effect, occupies 7.8% of the propeller disc. The small area of the ground effect region indicates that the experimental error is within an acceptable range. (2) Although the experimental stand is placed vertically, the height from the ground is only 30 cm, so there is still certain ground effect interference, but the influence can be neglected [27]. (3) Since the motor has mechanical loss, thus affecting the power measurement value of the DCP motor, the voltage and current values of the power supply cabinet at different speeds when the motor is idle are measured first, and then the propeller is installed for measurement. The difference is obtained by subtracting the two sets of values, which becomes the final power experimental data, as in Equation (10).

$$P_{text} = IU - I_e U_e \tag{10}$$

*3.2. Time-Step Sensitivity Test*

The lift and power consumption of the DCP in hovering state at different speeds are calculated. Since the DCP calculation has a complex flow field, the time step has a large influence on the calculation results. Thus, time-step analysis was carried out to eliminate the influence of time steps. In the *CFD* simulations, the time step is set as 0.0001 s (1.5°), 0.0002 s (3°), and 0.0004 s (6°) at 2500 RPM, and the data of 180° are selected. The calculation results are shown in Figure 9. It can be seen that when the time step is less than 0.0002 s, the results are close, so the time step is set as 0.0002 s in this study. Through the *CFD* method, the aerodynamic performance of the DCP was calculated, and the lift *L* and torque *Q* of the DCP in hovering were obtained. The *CFD* power ($P_{CFD}$) was calculated by Formula (8), where $\omega$ is the angular velocity.

$$P_{CFD} = Q * \omega \tag{11}$$

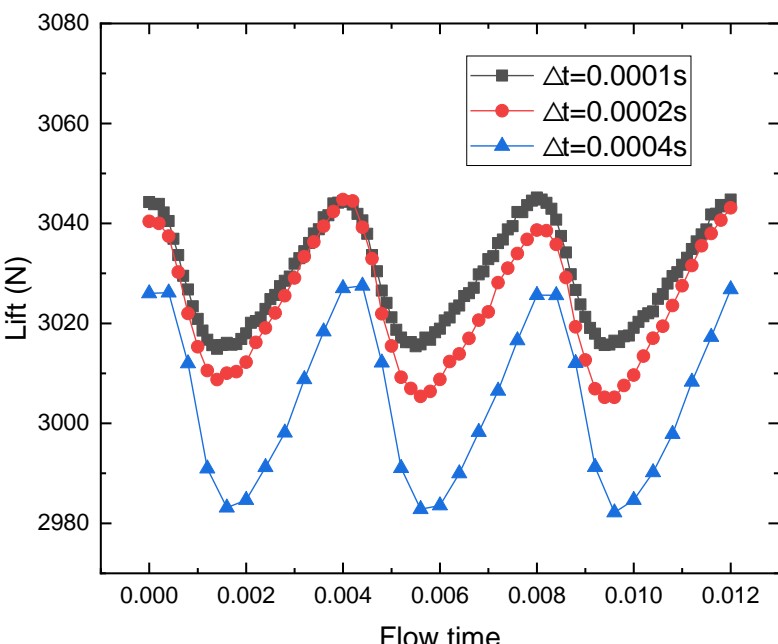

**Figure 9.** Lift history of coaxial propellers versus time steps.

### *3.3. Method Validation*

The experiments measured the lift, voltage, and current of the DCP at speeds from 500 rpm to 2500 rpm, and finally, the experimental values of lift and power consumption generated by the propeller were obtained. The above *CFD* method was used to calculate the same working conditions and models, and the experimental results and *CFD* calculation results are shown in Figure 10. It can be seen that: (1) The lift of the DCP is proportional to the quadratic of the rotational speed, the power consumption is proportional to the cubic of the rotational speed, and this test result can provide support for the aerodynamic modeling of the DCP. (2) When the test speed reaches 2510 RPM, the lift of the DCP reaches 3000 N, and the power is 79 kW. (3) Comparison with the experimental values is shown in Figure 10, which reveals that the calculated lift and power values are slightly smaller than the experimental values, and the errors are less than 5%.

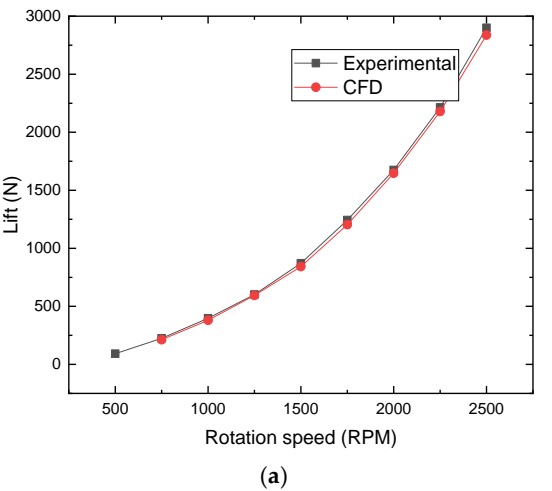

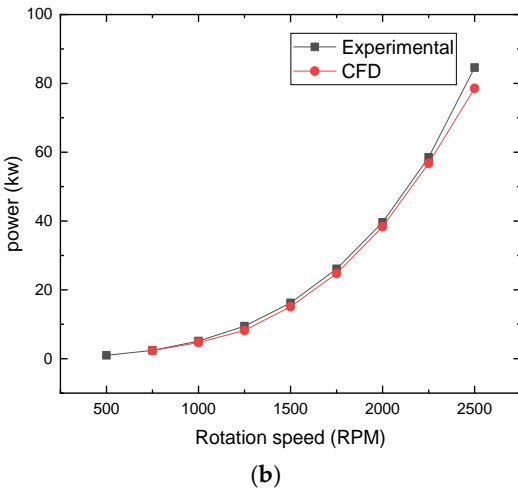

**(a)**　　　　　　　　　　　　　　　　　　　　　　　**(b)**

**Figure 10.** Comparison of *CFD* calculated values and experimental values: (**a**) relationship between lift and rotation speed; (**b**) relationship between power and rotation speed.

Several factors cause the errors. Firstly, there is a certain error in manufacturing the experimental DCP system, so some parts may have deformation, leading to the deviation of the experimental results from the numerical results; secondly, the *CFD* calculation model does not consider the influence of the propeller hubs and the motors, resulting in some deviation between the calculation results and the experimental results, but the motors and the propeller hubs are located at the root of the blades, so the aerodynamic influence is not strong; finally, there are complex flow fields inside the DCP such as blade tip vortex and airflow separation at the inner wall of the culvert, and there is interference between them, so the *CFD* calculation is difficult and there is a certain error of its own. Overall, the *CFD* method established in this paper is applicable to the simulation of the flow field of the DCP system.

## 4. Analysis and Discussion on Airflow Separation of DCP

### 4.1. Influence of Tip Clearance on Aerodynamic Characteristics

The formation of the tip vortex within the DCP, which directly affects the aerodynamic performance of the DCP, is mainly determined by the size of the tip clearance. In this paper, the DCP with a tip clearance of 0.336% (5 mm) is taken as the base model; other models obtain different tip clearances only by changing the diameter of the inner wall of the duct at a step of 2.5 mm. Hence, they have clearances of 0.503% (7.5 mm), 0.671% (10 mm), 0.839% (12.5 mm), 1.007% (15 mm), 1.174% (17.5 mm), and 1.342% (20 mm). Compared with the diameter of the duct inner wall, the tip clearance is very small, so the effect of the change in duct inner wall diameter due to the change in tip clearance on the DCP aerodynamics is neglected.

Figure 8 shows the variation of the DCP lift, torque, and power load with the tip clearance ratio for a DCP speed of 2500 RPM. Lift/power characterizes aerodynamic efficiency, and the duct lift factor characterizes the percentage of duct lift to total lift. From Figure 11, it can be seen that (1) the total lift, duct lift factor, and lift/power of the DCP all decrease as the tip clearance ratio increases; the lift and torque of the upper and lower propellers do not change much, where when the tip clearance ratio increased from 0.336% to 1.342%, the total lift and aerodynamic efficiency both decreased by about 11.3%. (2) With an increase in tip clearance, the air intake of the upper propeller is increased, and the lift of the upper propeller gradually increases. When it increases to the point that the duct stops functioning, the lift of the upper propeller does not increase anymore. In the meanwhile, the lift of the lower propeller will increase slowly at first, but when the clearance reaches 1.0007%, the aerodynamic interference of the upper propeller to the lower propeller plays a major role, so the growth trend of the lower propeller lift is smaller than the upper propeller. At this time, the lower propeller generates a smaller lift than the upper propeller. (3) The airflow environment at the duct mouth is complex, with an obvious unsteady phenomenon. The maximum velocity of the flow is located at the duct inner wall; with an increase in tip clearance, the tip vortex strength increases. The generated tip vortex blocks the airflow in the duct to some extent. The flow velocity at the duct lip decreases, resulting in the dropping of the duct lift. Therefore, the smaller the tip clearance ratio, the smaller the total pressure loss when the propeller unloads to the duct due to the better closure effect, which is the reason why the tip clearance needs to be reduced as much as possible when it is allowed [28].

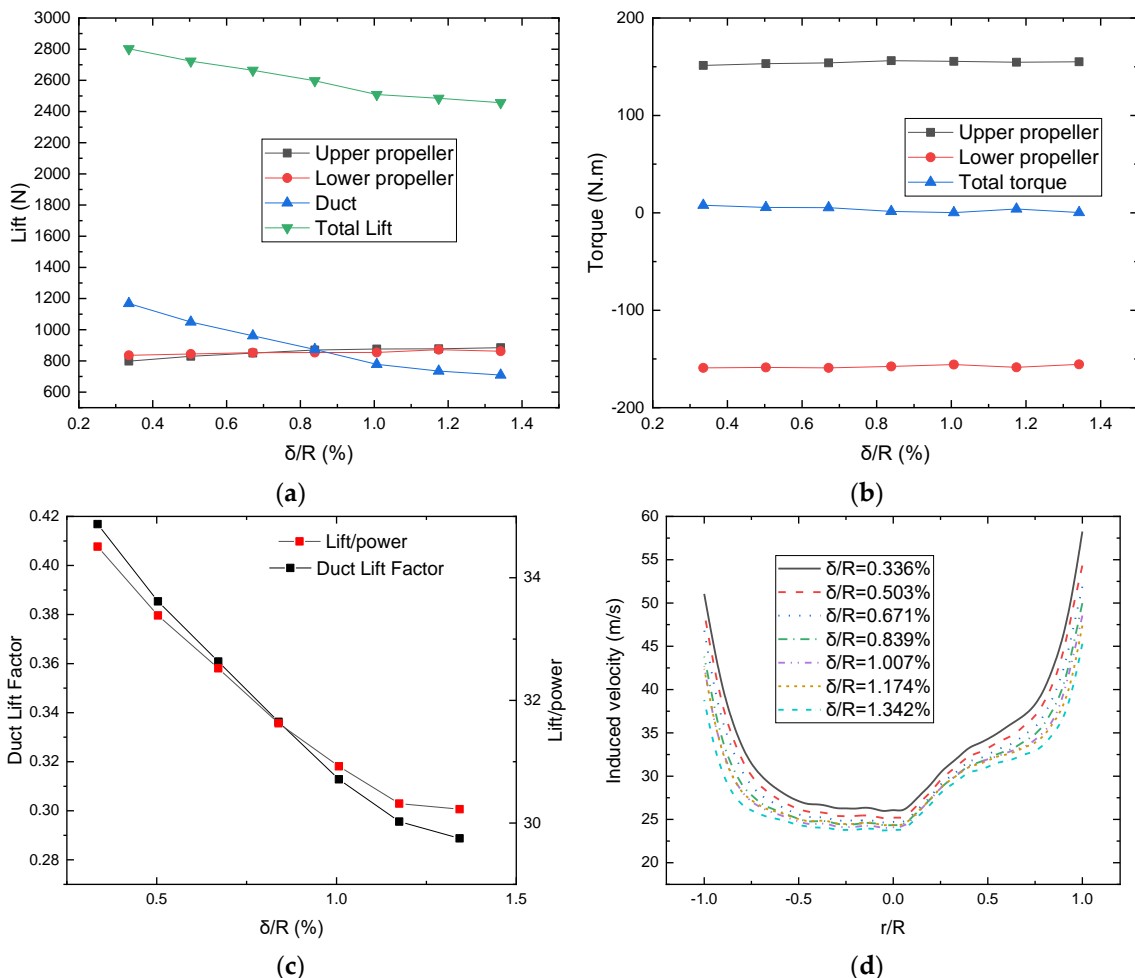

**Figure 11.** Relationship between aerodynamic characteristics and tip clearance ratio: (**a**) relationship between lift and tip clearance ratio; (**b**) relationship between torque and tip clearance ratio; (**c**) aerodynamic efficiency and duct lift factor of DCP; (**d**) induced velocity of duct lip's cross-section.

*4.2. Phenomenon and Mechanisms of Airflow Separation*

The tip clearance has a significant effect on the aerodynamic performance of the DCP, so it is necessary to understand the mechanism of its effect. Figure 12 shows the cross-sectional streamlines, induced velocity field, and vorticity fields for different clearance ratios. It can be seen that the airflow separation phenomena in the clearance flow field and the duct wall are very different for the four clearance ratios. The main phenomena are as follows: (1) Due to the existence of tip clearance, the tip vortex will be generated in the area between the propellers' tips and ducts, and the tip vortex generated by the lower propeller is stronger than that of the upper propeller under the same clearance. The airflow separation occurs in both the linear and diffusion sections of the inner wall of the duct, but it is more obvious in the diffusion section. (2) Under the tip clearance ratio of 0.336%, the flow at the tip clearance is less blocked and the tip vortex is smaller, and the airflow separation along the duct wall is smaller; as the tip clearance increases, the tip vortex at the tip of the propeller becomes stronger and stronger, and the flow is blocked and the energy loss is larger. As the tip clearance increases, the tip vortex becomes stronger, the flow is blocked, the energy loss is greater, the quality of airflow decreases, and the airflow separation along the duct wall becomes more and more significant, squeezing the airflow space inside the duct; when the tip clearance ratio is 1.342%, the upper and lower propellers have a larger leakage vortex at the tip of the propeller.

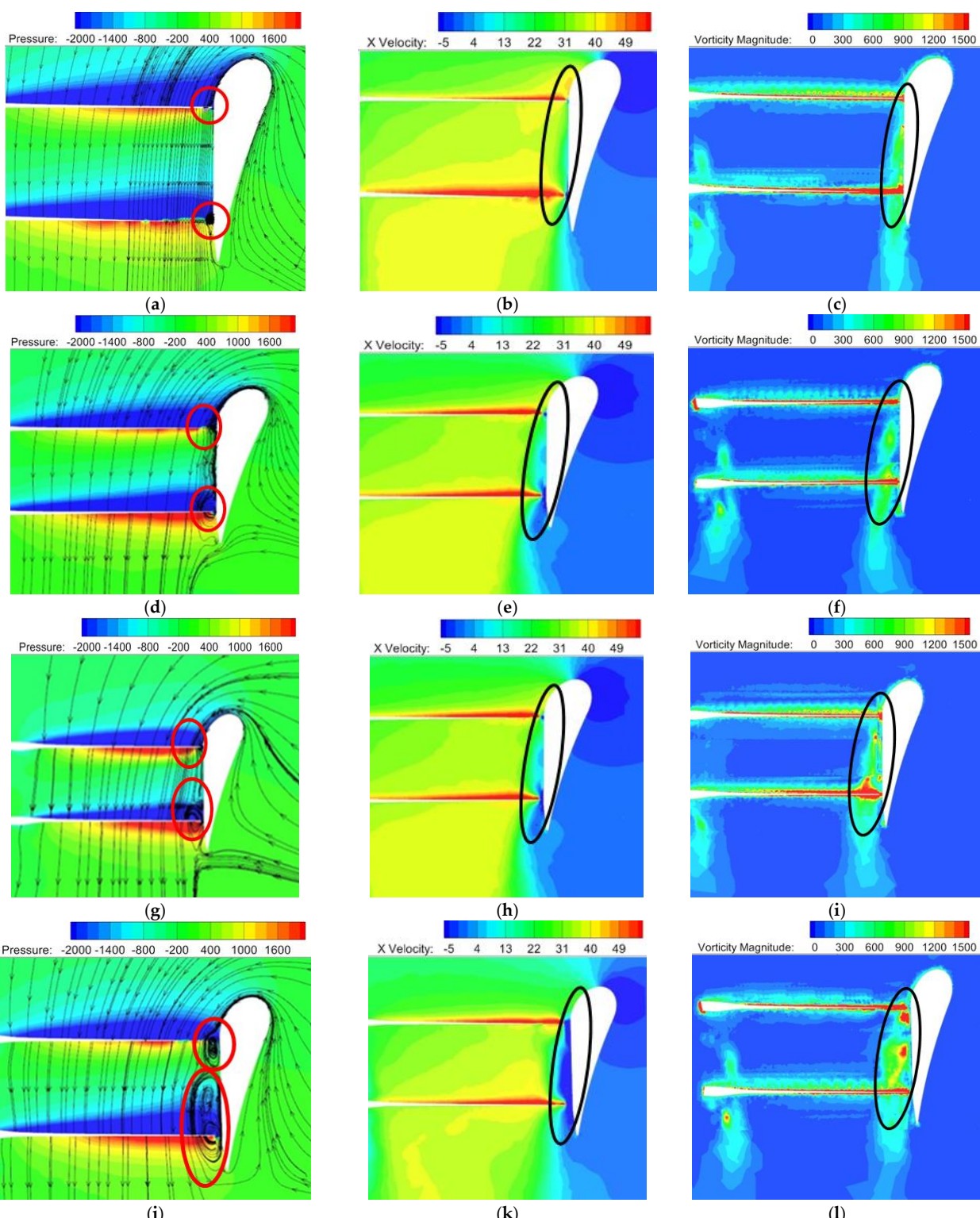

**Figure 12.** Flow field of flow separation, the red circles indicat the tip vortex, the balck circles indicat the flow separation: (**a**) streamline diagram and pressure field ($\delta/R$ = 0.336%); (**b**) axial induced velocity field ($\delta/R$ = 0.336%); (**c**) vorticity field ($\delta/R$ = 0.336%); (**d**) streamline diagram and pressure field ($\delta/R$ = 0.671%); (**e**) axial induced velocity field ($\delta/R$ = 0.671%); (**f**) Vorticity field ($\delta/R$ = 0.671%); (**g**) streamline diagram and pressure field ($\delta/R$ = 1.007%); (**h**) axial induced velocity field ($\delta/R$ = 1.007%); (**i**) vorticity field ($\delta/R$ =1.007%); (**j**) streamline diagram and pressure field ($\delta/R$ = 1.342%); (**k**) axial induced velocity field ($\delta/R$ = 1.342%); (**l**) vorticity field ($\delta/R$ = 1.342%).

Mechanisms of generation of the tip vortex in the duct and of its influence on aerodynamic characteristics: The tip vortex is generated due to the pressure difference between the upper and lower surfaces, and the ring bracket effect of the duct can significantly reduce the intensity of the tip vortex and reduce the energy loss. However, the tip clearance inevitably exists; thus, generation of the tip vortex is also difficult to avoid. The larger the tip clearance, the greater the intensity of the tip vortex generated. With an increase in the tip clearance ratio, the leakage vortex loss in the tip clearance area increases, the improvement effect of the duct on the downstream of the propeller slipstream is reduced, and the unloading effect of the duct on the propellers is also reduced. Specifically, on the one hand, the effective angle of attack of the propeller near the tip of the blade element is reduced, thus reducing the propeller lift and aerodynamic efficiency. On the other hand, the airflow speed at the duct lip is decreased, and thus the negative pressure decreases, resulting in the reduction of duct lift, which leads to a decrease in aerodynamic efficiency with an increase in the tip clearance ratio. The lower propeller generates a stronger tip vortex than the upper propeller because at a clearance of 0.336% (5 mm), the lift generated by the lower propeller is greater than that of the upper propeller due to the suction effect of the upper propeller; however, with an increase in the tip clearance, the lift loss of the lower propeller increases. Combining the interference of the upper propeller wake, the lift generated by the upper propeller will be greater than that of the lower propeller. From Figure 13, it can be seen that the tip vortex affects not only the tip of the propeller, but also the whole intersection area between the propellers and the duct wall during the high-speed rotation of the propeller, which provides opportunities for flow separation in the duct.

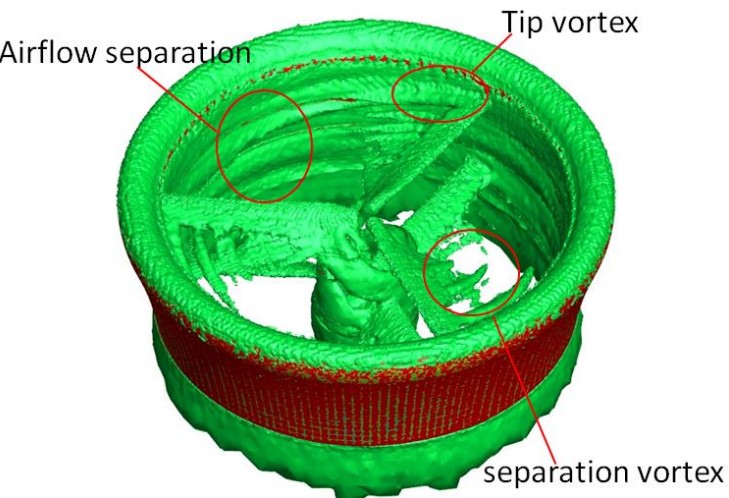

**Figure 13.** Vorticity diagram.

Mechanism of airflow separation in the line section and diffusion section: Usually, an obvious wake contraction effect can be observed when the propellers rotate. Although the duct inner wall can suppress wake contraction and improve the airflow state below the upper propeller and lower propeller in the DCP, factors that may result in wake contraction still exist due to the existence of the tip clearance. When the tip vortex in the duct moves downward along the inner wall of the duct with the axial airflow, airflow separation is induced together by the viscous effect of the duct wall and the wake shrinkage, which blocks the inner wall of the duct, reduces the effective inner diameter of the duct, and lowers the airflow through the duct. As for the diffusion port below the lower propeller, airflow separation is more likely to occur due to flow expansion and the reverse pressure gradient at the port, resulting in a back flow of air, which directly leads to airflow separation at the duct diffusion port. Therefore, airflow separation is more likely to occur in the diffusion port than in the straight section for the DCP.

In sum, tip clearance has a significant influence on the aerodynamic performance of the DCP, which is mainly due to the formation of the tip vortex, airflow separation in the straight section, and the diffusion section of the duct. Firstly, the tip vortex and airflow separation increase energy dissipation; secondly, the tip vortex blocks the inner wall of the duct, reducing the effective inner diameter of the duct, decreasing the airflow through the duct, and thus affecting the aerodynamic efficiency; finally, the role of the duct is weakened, and the wake is contracted, which increases the induced velocity and thus the induced power loss.

### 4.3. Suppression of DCP Airflow Separation

According to the analysis in Section 3, in order to reduce the negative influence of the tip vortex and airflow separation on aerodynamic performance of the DCP, and also the design processing requirements of the tip clearance, some active flow control methods to suppress the tip vortex and airflow separation were proposed [29,30], and in this paper, adding different types of VRR models to the inner wall of the duct are proposed to improve the airflow field inside the duct. In this paper, the effect of adding a 5 mm VRR in the straight section and diffusion section on the aerodynamic characteristics of the DCP is studied. The inner wall of the duct after adding the VRR is shown in Figure 14. Table 2 shows the calculation results after adding the VRR in different sections. It can be seen that the total lift of the DCP is reduced by 5.8% by adding the VRR in the straight section of the duct. Under the effect of induced velocity, the prominent VRR generates negative lift due to the blocking effect, and thus reduces the lift generated by the duct. Adding the VRR in the diffusion section increases the total lift of the DCP by 5.1%, and the main contribution to the lift increment is the duct. As for the power consumption, it remains basically the same after adding the VRR.

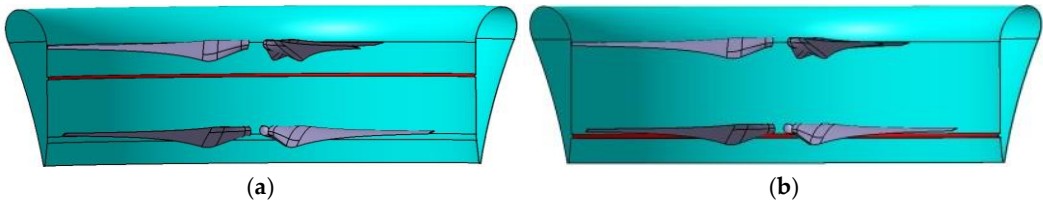

(**a**)  (**b**)

**Figure 14.** Results after adding VRR: (**a**) VRR in straight section; (**b**) VRR in diffusion section.

**Table 2.** Lift and torque distribution of different components.

| VRR Location | Lift of the Upper Propeller (N) | Lift of the Lower Propeller (N) | Lift of Duct (N) | Duct Lift Factor | Power (kw) |
|---|---|---|---|---|---|
| Without VRR | 798.4 | 835.8 | 1168.2 | 0.417 | 81.2 |
| VRR in straight section | 766.8 | 876.7 | 996.4 | 0.377 | 81.3 |
| VRR in diffusion section | 842.3 | 862.5 | 1240.9 | 0.421 | 81.6 |

As for the working mechanisms of the VRR, it can be seen from Figure 15a–c that when the VRR is added to the straight section, the airflow separation is improved. It works as follows: the VRR blocks the tip vortex of the upper propeller from axial motion, and velocity of the vortex is suppressed on the windward side of the VRR, and the mainstream is reattached to the inner wall of the duct after reaching the leeward side of the VRR. Since the reattached fluid in the region near the wall has high energy, it can overcome the airflow separation caused by the inverse pressure gradient of airflow contraction. Though the airflow in the straight section is improved, the axially induced velocity creates a high-pressure area on the windward side of the VRR, which reduces the role of the duct and thus aerodynamic efficiency. From Figure 15d–f, it can be seen that adding the VRR in the diffusion section serves to improve aerodynamic efficiency in two ways. On the one hand,

it hinders the formation of the tip vortex of the lower propeller and reduces the strength of the tip vortex. On the other hand, it overcomes the contraction of the airflow and the inverse pressure gradient of diffusion along the mouth of the duct, thus hindering the return flow of the airflow and weakening the airflow separation. Collectively, adding the VRR to the diffusion section of the DCP can improve the airflow environment at the inner wall of the duct, weaken the airflow separation phenomenon, and improve aerodynamic efficiency by 5.1%.

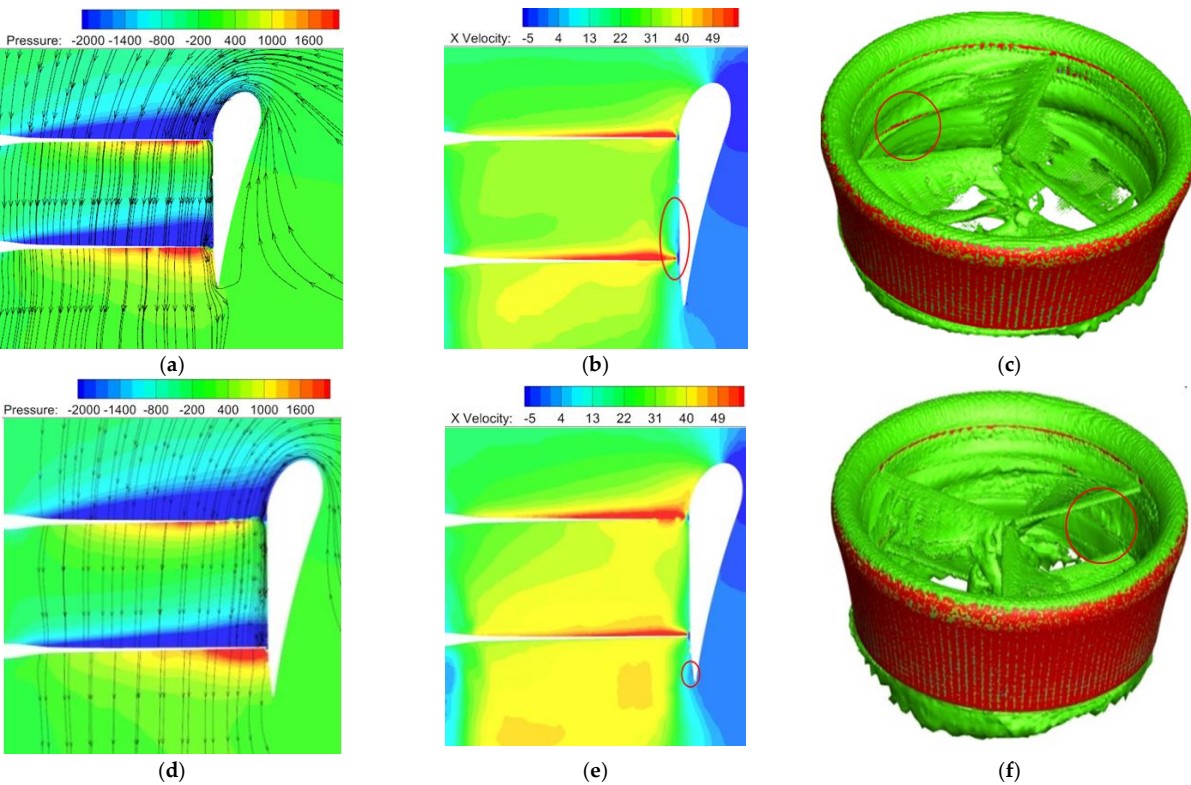

**Figure 15.** Flow field after adding VRR: (**a**) streamline diagram and pressure field (straight section); (**b**) axial induced velocity field (straight section); (**c**) vorticity diagram (straight section); (**d**) streamline diagram and pressure field (diffusion section); (**e**) axial induced velocity field (diffusion section); (**f**) vorticity diagram (diffusion section).

## 5. Conclusions

In this paper, a numerical simulation method based on the sliding mesh technique that can simulate the unsteady flow field of the DCP is established, a set of DCP lift systems with a diameter of 1.78 m is designed and fabricated, and an experimental platform is built to verify the confidence of the *CFD* method with the experimental data. The airflow separation mechanism and suppression method of the inner wall of the duct is studied. The main conclusions are as follows.

(1)  The effect of tip clearance on the aerodynamic performance of the DCP is relatively significant. When the tip clearance ratio increases from 0.336% to 1.342%, the total lift and aerodynamic efficiency both decrease by about 11.3%. The effects mainly lie in the formation of the tip vortex, airflow separation in the straight section, and diffusion section of the inner wall of the duct. Firstly, the tip vortex and airflow separation increase energy dissipation; secondly, the vortex blocks the inner wall of the duct, reduces the effective inner diameter, and decreases the airflow through the duct; finally, the role of the duct is weakened, and the wake is contracted, which increases the induced velocity and thus the induced power loss.

(2) The mechanism of airflow separation in the straight section and the diffusion outlet of the duct: when the tip vortex in the duct moves downward along the inner wall with the axial airflow, airflow separation occurs, induced by the viscous effect of the duct wall and the contraction of the wake together, which blocks the inner wall area of the duct, reducing the effective inner diameter and lowering the airflow through the duct. As for the diffusion port below the lower propeller, airflow separation is more likely to occur due to flow expansion and the reverse pressure gradient at the port, resulting in air backflow, which directly leads to airflow separation at the duct diffusion port. Therefore, airflow separation is more likely to occur in the diffusion port than in the straight section for the DCP.

(3) Adding the VRR to the inner wall diffusion section of the duct can effectively suppress the occurrence of the tip vortex and airflow separation and improve the airflow flow quality inside the duct, thus improving aerodynamic efficiency by 5.1%.

**Author Contributions:** Author Contributions: Conceptualization, J.W., R.C. and J.L.; methodology, J.W. and R.C.; software, J.W. and J.L.; validation, J.W. and R.C.; formal analysis, J.W. and R.C.; investigation, J.W. and R.C.; resources, J.W.; data curation, J.W., R.C. and J.L.; writing—review and editing, J.W., J.L. and R.C.; visualization, J.W. and R.C.; supervision, J.W. and R.C.; project administration, J.W.; funding acquisition, R.C. All authors have read and agreed to the published version of the manuscript.

**Funding:** This research was funded by the National Natural Science Foundation of China, grant number No. 11672128.

**Institutional Review Board Statement:** Not applicable.

**Informed Consent Statement:** Informed consent was obtained from all subjects involved in the study.

**Data Availability Statement:** The data used to support the findings of this study are available from the corresponding author upon request.

**Acknowledgments:** Thanks for the Project Funded by the Priority Academic Program Development of Jiangsu Higher Education Institutions.

**Conflicts of Interest:** The authors declare that there are no conflict of interest.

## Nomenclature

| | |
|---|---|
| $\Phi$ | Generalized flux |
| $\rho$ | Air density |
| $V$ | Velocity vector |
| $\Gamma_\Phi$ | Diffusion coefficient |
| $S_\phi$ | Source term |
| V0 | Free flow velocity |
| R | Radius of the rotor |
| $\nu_h$ | Induced velocity |
| y+ | Dimensionless wall distance |
| $L_{Newmesh}$ | Lift of the new grid |
| $L_{Oldmesh}$ | Lift of the last grid |
| $P_{CFD}$ | *CFD* power |
| $P_{text}$ | Text power |
| UAM | Urban air mobility |
| eVTOL | Electrically driven vertical take-off and landing |
| DCP | Ducted coaxial propeller |
| VRR | Vortex restrain ring |
| FCP | Free coaxial propeller |
| DSP | Ducted single propeller |
| *CFD* | Computational fluid dynamics |
| ESC | Electronic stability controller |
| SST | Shear stress transport |

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
