# Peer review of "Experimental and Numerical Studies on the Effect of Airflow Separation Suppression on Aerodynamic Performance of a Ducted Coaxial Propeller in Hovering"

_aerospace, doi:10.3390/aerospace10010011_

Round 1
Reviewer 2 Report
This is a very interesting article. The papers showed a numerical simulation of airflow separation of the ducted coaxial propeller in hovering. The introduction presents a review of the literature on experimental and numerical research on the Ducted coaxial propeller. The Methods section describes the model, meshing and numerical methods used. Then, the results of model validation and the analysis of the results along with their discussion were presented. In my opinion, the discussion on the results should be supplemented with reference to the results of other authors. The conclusions are supported by research results. I have a few comments:
1. No information about the size of the mesh used in Fluent CFD calculations.
2. There is no information about the results of the measurements.
3. No information about the error analysis of the study method. Could you write some information about uncertainty in your study? Do you consider the impact of the uncertainty on the results of your work?
4. In the discussion of the results, there are no references to studies of similar systems by other authors.
Round 2
Reviewer 1 Report
Dear authors,
I appreciate your thorough editing of the paper. I believe your manuscript is interesting and scientifically sound and can be accepted for publication. I wish you rewarding scientific studies and every success with your future publications.
Reviewer 2 Report
Thank you for explanation. I have no more comments